# Influenza vaccine hesitancy among healthcare workers in a Northeastern province in Thailand: Findings of a cross-sectional survey

**Manash Shrestha[1], Penchan Pradubmook Sherer[1], Seung Chun Paek[1], Kriengkrai Prasert[2,3], Sutthinan Chawalchitiporn[4], Prabda Praphasiri[3]***

**1** Department of Society and Health, Faculty of Social Sciences and Humanities, Mahidol University, Nakhon Pathom, Thailand, **2** Nakhon Phanom Hospital, Nakhon Phanom, Thailand, **3** Faculty of Public Health, Kasetsart University, Sakon Nakhon, Thailand, **4** Epidemiology Department, Faculty of Medicine, Thammasat University, Bangkok, Thailand

* marxprabda2510@gmail.com, Prabda.p@ku.ac.th

## Abstract

### Background

Healthcare workers (HCWs) are an important target group for influenza vaccination due to their increased risk of infection. However, their uptake remains a challenge. This study aimed to identify and measure influenza vaccine hesitancy among HCWs in Nakhon Phanom province, Thailand.

### Methods

A representative cross-sectional survey was conducted during August–September 2020, among 350 HCWs at six hospitals in the province selected by a two-stage cluster sampling using a self-administered questionnaire. HCWs who either delayed getting influenza vaccines, accepted the vaccines but were unsure, or refused the vaccine with doubts were categorized as hesitant. HCWs who accepted the influenza vaccine without any doubts were classified as non-hesitant. Determinants of vaccine hesitancy were identified by a multivariable logistic regression analysis.

### Results

A total of 338 participants (97%) filled the questionnaires. The mean age of the participants was 37.2 years. Most participants were female (280; 83%), nurses (136; 40%), working at district hospitals (238; 71%), with bachelor's degree (223; 66%), and without any pre-existing chronic medical conditions (264; 78%). Influenza vaccine hesitancy was evident among nearly 60% of the participants (197/338), who had varying patterns of hesitancy. Significant factors of influenza vaccine hesitancy were found to be age above 50 years (adjusted odds ratio [aOR] 3.2, 95% CI 1.3–8.5), fair knowledge of influenza and vaccination (aOR 0.4, 95% CI 0.2–0.8), and negative influence of other HCW (High level–aOR 2.3, 95% CI 1.1–4.8; Moderate level- aOR 2.1, 95% CI 1.1–4.4).

**Data Availability Statement:** All relevant data are within the manuscript and its Supporting Information files.

**Funding:** The author(s) received no specific funding for this work.

**Competing interests:** The authors have declared that no competing interests exist.

## Conclusion

Influenza vaccine hesitancy was highly prevalent among the Thai HCWs in Nakhon Phanom province. Imparting updated information to the HCW, in combination with positive guidance from influential HCWs in the hospital, may help reduce hesitancy. These data may be useful to the National Immunization Program to design appropriate approaches to target hesitant HCWs in Thailand to improve influenza vaccine coverage.

## Introduction

Healthcare workers (HCWs) are recommended for seasonal influenza vaccines as they are at a higher risk of contracting influenza and inadvertently transmitting the virus to high-risk and vulnerable patients who may develop severe complications [1–5]. However, despite several recommendations, influenza vaccine uptake among HCWs has remained a challenge globally [6–8]. Many studies indicate that despite vaccine availability, some HCWs may be hesitant about vaccines, especially influenza vaccines, including when considering the vaccines for their children or their patients [9–11].

While the anti-vaccination movement is not a recent development [12], more people are becoming skeptical of vaccines, such that vaccine hesitancy is now considered a major threat to global health [13]. As HCW recommendations are highly influential for patients [14, 15], acceptance of vaccination by HCWs themselves is, therefore, also an important determinant of influenza vaccine uptake in the populations that they serve [16, 17]. Addressing the concerns of HCWs may be strategically beneficial in reducing vaccine hesitancy among the general public as well [18].

Defined as "a delay in acceptance or refusal of vaccines despite availability of vaccination services" by the World Health Organization (WHO)'s Strategic Advisory Group of Experts (SAGE), vaccine hesitancy is a behavioral phenomenon that broadens the scope of study of vaccination decision-making [19]. Vaccine-hesitant people are seen in a spectrum between those who accept vaccination without any doubts and those who reject vaccination at all costs. People may either accept vaccines without hesitation, accept them with hesitation, or reject them with or without hesitation [20]. Vaccine-hesitant individuals might receive all recommended vaccines on time but still harbor significant doubts [21]. Thus, vaccine hesitancy is better understood as a state of indecision and reluctance [22]. This view frames vaccine hesitancy as involving 'doubts', 'concerns', and 'reluctance' about vaccination. Many scholars similarly argue that vaccine hesitancy is more a psychological state than a behavioral one [20, 23–25].

In their "3C" model, SAGE identified "complacency", "inconvenience in accessing vaccines", and "lack of confidence" as three key reasons for this hesitancy [19]. However, this phenomenon is dependent on the context and specific to vaccines. For example, the hesitancy to influenza vaccine may be different than to other vaccines. Therefore, SAGE supplemented the "3C" model with a vaccine hesitancy determinant matrix that included contextual, individual and group, and vaccine/vaccination-specific influences [19]. Although Paterson et al. adapted this matrix for HCW and added a few more determinants [16], it has been informed largely by research from high-income countries. Evidence from middle-income countries like Thailand has remained minimal.

In Thailand, HCWs were the first group to be recommended for yearly influenza vaccination by the Ministry of Public Health (MOPH) since 2004 [26]. Vaccination rates among Thai

HCWs were self-reported to be as high as 89% in 2009 [27]. However, in 2018, regional reports showed that HCW influenza vaccine coverage had plummeted to below 30% in some regions (Department of Medical Services [DMS], MOPH) (S1 File). Although little is known about Thai HCW's attitudes towards influenza vaccines, there are some disconcerting pieces of evidence. A study revealed that only 60% of Thai HCWs were willing to receive the pandemic influenza A(H1N1) vaccine [28]. In 2013, more than one-third of the Thai physicians working in ante-natal care clinics were found to have doubts about influenza vaccine's safety and effectiveness, and only 25% of them routinely recommended influenza vaccines to pregnant women [29]. Another study reported that the seasonal influenza vaccine acceptance rate among medical professionals at a tertiary hospital in Bangkok was 65.4% [30]. Being the most informed group of people about the benefits of vaccines and having more access to vaccine than others, moderate acceptance and lower recommendations of influenza vaccines by Thai HCW may indicate deeper-lying cultural or personal concerns that might relate to vaccine hesitancy.

Despite being the study site of many influenza vaccine-related research in Thailand in the recent past that have actively partnered with local HCWs [31–33], the HCW uptake rate in Nakhon Phanom province was only 28% in 2018 (S1 File). Low uptake of influenza vaccines among HCWs is particularly remarkable in this border northeastern province as it is considered a "model" province for influenza vaccine research in Thailand, where vaccination rates are supposed to be higher. As Thailand's influenza vaccination policies were largely shaped by the active surveillance data from this province [34–36], we found it important to evaluate if the HCWs in the province were themselves hesitant to influenza vaccines. Therefore, the objectives of this cross-sectional survey were to identify and measure influenza vaccine hesitancy among HCWs in Nakhon Phanom province.

## Materials and methods

### Study design and setting

This cross-sectional survey was part of a mixed-methods study on influenza vaccine hesitancy among healthcare workers in Nakhon Phanom province, which is located in the plateau region of northeastern Thailand, approximately 735 km from Bangkok. The province borders Lao PDR and has a total area of 5,528.88 sq. km. The province consists of 12 districts, with Muang district as the capital, along with 97 sub-districts and 1,123 villages [34]. There are 14 public hospitals in the province: one provincial general hospital, 11 community hospitals, a psychiatric hospital, and a military hospital. From 2003 to 2015, there was active surveillance of respiratory illnesses in the public hospitals of this province [34–36].

### Study population

Thai HCWs of Nakhon Phanom province were the targeted population of the study. Although the WHO defines HCW as "all people engaged in actions whose primary intent is to enhance health" [37], we limited HCW's definition to those who were currently working in public hospital settings in Nakhon Phanom for operational feasibility. As per a report of DMS, 1,863 registered HCWs were working in the province in 2018 (DMS, MOPH). The HCWs were broadly categorized into two groups according to the amount of patients' contact/exposure and as grouped by WHO [38]: a) *Health service providers* who delivered health services and came in direct contact with patients and substances such as patients' blood and other specimens, and b) *Health management and support staff* who assisted in the functioning of the health system without directly providing health services and as such may or may not have had direct contact with the patients. According to the nature of their work, health service providers were further

sub-categorized into six groups—medical doctors, dentists, nurses, pharmacists, public health officers, and other service providers [38].

### Participant selection criteria

Thai HCWs of both sexes (male and female) aged above 18 years, having a good command of the written and spoken Thai language, and currently working at a public hospital in Nakhon Phanom province were included in the study. HCWs who were not present in the hospital at the time of data collection and who were not willing to participate in the study voluntarily were excluded.

### Sample size and sampling technique

As there was no prior estimate of influenza vaccine hesitancy among Thai HCWs, a conservative proportion of 50% was taken to calculate the sample size for the survey. A sample size of 350 was calculated using the single proportion formula and a finite population correction factor in OpenEpi (www.openepi.com), assuming a Type I error of 5%, absolute precision of 7.5 percentage points, and a design effect of 2 given the cluster sampling used for the population size of 1,863 HCW in Nakhon Phanom after inflating 10% for refusal and absences.

HCWs were recruited in the survey using a two-stage cluster sampling. In the first stage, hospitals were selected as clusters to recruit 50 HCWs in each cluster (i.e., seven clusters in total). The hospitals were chosen by a systematic random sampling using probability proportional to size (PPS) of the number of HCWs. As Nakhon Phanom Provincial hospital was a large hospital, it got selected twice. Therefore, 100 HCWs were recruited from that particular hospital, and 50 HCWs were selected from five other hospitals. In the second stage, HCWs were chosen using stratified random sampling according to their professional categories and proportional distribution in each hospital. Microsoft Excel sheets were used to generate the random numbers for the sampling in both stages, as described above.

### Survey instrument and data collection

The survey questions were modified from the sample questions of SAGE's vaccine hesitancy determinant matrix for the Thai context to measure influenza vaccine hesitancy and its three broad determinants: contextual or sociocultural, individual/group, and influenza vaccine-specific influences [16, 39]. Clear and concise language was used for ease of understanding and questions that were related were grouped to avoid jumping between unrelated topics. The participants were asked to indicate their level of agreement or disagreement with the items of the constructs on a 5-point Likert scale, with '1' meaning strongly disagree and '5' strongly agree. The questionnaire took around 30–35 minutes to complete. The study instrument was validated for use in the Thai context by another research project which used qualitative methods in the development of the questionnaire and had rigorous steps such as pilot testing (n = 30) for assessing feasibility, acceptability, and face validity, and a larger survey among a nationally representative sample of healthcare workers (n = 2352) for evaluating construct validity and reliability [40].

In this survey, the questionnaire was self-administered to the HCWs who were approached through designated focal points at the six hospitals in the province, identified by the sampling technique, from 1st August to 30th September 2020. A total of 350 questionnaires were distributed to the participants. The participants were assured that their responses would remain anonymous and confidential to reduce the pressure to conform to socially desirable responses. Clear instructions were provided on how to answer the questions, emphasizing the importance of honest and accurate responses.

## Variables and measurement

The dependent variable of the study, influenza vaccine hesitancy, was defined as either delaying getting influenza vaccines, accepting the vaccines but being unsure, or refusing the vaccine despite seeing its value. It was measured as a binary variable (yes/no). "Yes" meant those HCWs who displayed some form of hesitancy, and "No" meant HCWs who accepted the influenza vaccine without any doubts.

The independent variables included demographic and work characteristics. Participant's age and years of experience as HCW were first measured as continuous variables and then converted to appropriate categories. Categorical variables were sex (binary—male/female), religion, level of the hospital (Provincial/District), HCW type (Health service providers/Health management and support staff), daily exposure to patients (<1 meter/≥1 meter/no contact), education level, and self-reported presence of any pre-existing medical conditions (yes/no). Participants' knowledge of influenza and vaccination was assessed using seven items that were adopted as a subset of a questionnaire from a previous study among Thai physicians [29], and then grouped into three categories using a modification of Bloom's original cut-off points, such that six or more correct answers (i.e. ≥80%) represented "good knowledge", four to five correct answers (i.e. 60–79%) meant "fair knowledge" and three or less correct answers (i.e. <60%) were considered as "needed improvement".

The determinants of vaccine hesitancy were measured as constructs (i.e., composite variables). Contextual factors included the negative influence of media/social media (2 items), politics/policies (2 items), pharmaceutical industry influences (2 items), and the level of trust in the health system (2 items). Individual/group influences contained constructs of experience with past vaccination (3 items), beliefs and attitudes about influenza and vaccination (3 items), vaccination as a social norm (3 items), and influence of other healthcare professionals (2 items). Influenza vaccine/vaccination-specific issues included constructs of risk/benefit (perceived, heuristic) (4 items), risk of adverse events due to vaccination (3 items), fear of painful injections (2 items), and access to vaccines in the hospital (4 items) (S1 Table). Both positively and negatively worded statements were used, and the direction of negatively worded questions was reversed for data analysis. Items under each construct were averaged to create aggregated mean scores for the construct. The average scores were case-ranked and then used to divide the construct into three categories–high, moderate, and low.

## Data analysis

Descriptive statistics were first employed to provide a summary of the selected variables and study samples. The mean scores of vaccine hesitancy determinants were compared between hesitant and non-hesitant HCWs using independent t-tests. The associated factors of influenza vaccine hesitancy were identified by a series of binary logistic regression models. As the dependent variable (influenza vaccine hesitancy) was categorical, the logistic regression analysis was performed using non-hesitant HCW as the reference category. All variables showing at least some evidence of association (Type III p values <0.1 in the binary logistic regression models) were entered into a multivariable logistic regression model, adjusting hospital locations to account for clustering. Variables with significant adjusted odds ratio (aOR) in the final model were recognized as associated factors (p-value <0.05). All data analysis was conducted using SPSS version 20 (IBM Corp., Armonk, NY).

## Ethical considerations

Written informed consent was obtained from each participant before data collection. The study protocols and all related study documents were reviewed and approved by the Mahidol

University—Central Institutional Review Board (MU-CIRB) (Protocol number MU-CIRB 2020/138.1606). Permission for data collection was also received from the Provincial Chief Medical Officer, Nakhon Phanom Provincial Public Health Office (ref no. 78.02/ 06115).

## Results

### Response rate

Out of the total 350 survey questionnaires distributed, 338 filled questionnaires were received back from the participants, implying a response rate of 96.6% (Fig 1). The response rates varied at different hospitals, ranging from 80–100% (S2 Table).

### Participant characteristics

Of those 338 participants, the mean age of the participants was 37.2±9.7 years, with a minimum of 20 years and a maximum of 60 years. Almost all participants were Buddhists (333; 98.5%). A majority of the participants were female (280; 82.8%), nurses (136; 40.2%), from district hospitals (238; 70.7%), with bachelor's degree (223; 65.9%), and without any pre-existing chronic medical conditions (264; 78.1%) (Table 1). The participants had a mean work experience of 13.8±9.8 years as HCW, and around half of them reported having high daily exposure with patients in the proximity of less than 1 meter (172; 50.9%).

### Knowledge about influenza and vaccination

A high majority of HCWs were aware of the MOPH recommendation of influenza vaccination for them (302; 89.3%). However, most participants were misinformed about the influenza vaccines as only 52% correctly identified that the influenza vaccine may not work if it contained the wrong virus strains than the circulating ones, and only 30% firmly believed that the

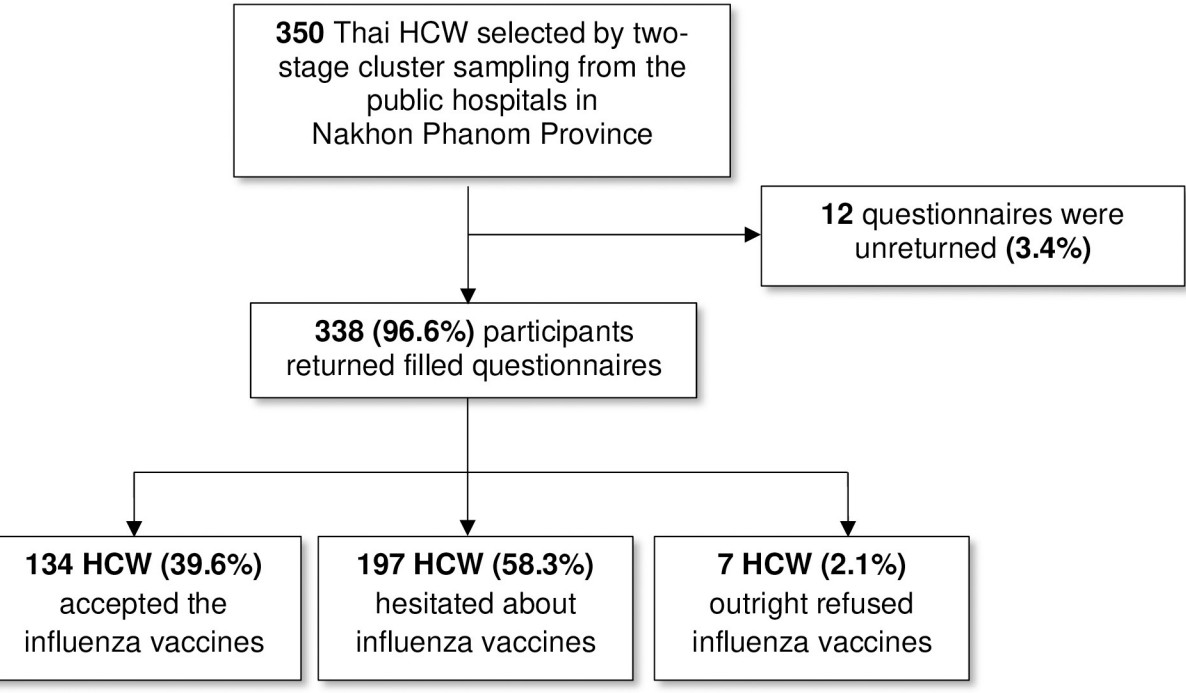

**Fig 1. Study flow and influenza vaccine hesitancy among healthcare workers in Nakhon Phanom Province, Thailand.**

**Table 1. Participant characteristics (n = 338).**

| Characteristic | Number | Percentage |
|---|---|---|
| Age group | | |
| ≤30 years | 102 | 30.2 |
| 31–40 years | 121 | 35.8 |
| 41–50 years | 72 | 21.3 |
| >50 years | 43 | 12.7 |
| *Mean (SD): 37.2 years (9.7); Min: 20, Max: 60* | | |
| Sex | | |
| Female | 280 | 82.8 |
| Male | 58 | 17.2 |
| Religion | | |
| Buddhism | 333 | 98.5 |
| Christianity | 5 | 1.5 |
| Level of hospital | | |
| Provincial | 100 | 29.6 |
| District | 238 | 70.4 |
| HCW type | | |
| Health service providers | 295 | 87.3 |
| Nurse | 136 | 40.2 |
| Pharmacist | 19 | 5.6 |
| Public health officer | 19 | 5.6 |
| Medical doctor | 16 | 4.7 |
| Dentist | 6 | 1.8 |
| Other medical personnel | 99 | 29.3 |
| *Hospital assistant staff* | *55* | *16.3* |
| *Patient assistants* | *27* | *8.0* |
| *Emergency medical technicians (EMT)/paramedic* | *7* | *2.1* |
| *Laboratory staff* | *7* | *2.1* |
| *Nurse aids* | *3* | *0.9* |
| Health management and support staff | 43 | 12.7 |
| Average daily exposure with patients | | |
| High (<1 meter) | 172 | 50.9 |
| Moderate (1 meter or more) | 106 | 31.4 |
| Low (no direct patient contact) | 60 | 17.8 |
| Work experience as HCW | | |
| ≤5 years | 74 | 21.9 |
| >5 years to <20 years | 166 | 49.1 |
| ≥20 years | 98 | 29.0 |
| *Mean (SD): 13.8 years (9.8); Min:1, Max: 37* | | |
| Highest educational attainment | | |
| High school or below | 36 | 10.7 |
| Diploma or vocational college | 43 | 12.7 |
| Bachelor's degree | 223 | 65.9 |
| Master's degree or higher | 36 | 10.7 |
| Pre-existing chronic medical condition | | |
| Yes | 74 | 21.9 |
| No | 264 | 78.1 |

Abbreviations: SD, Standard deviation; Min, Minimum; Max, Maximum

**Table 2. Participants' correct responses to each item of knowledge on influenza and vaccination (n = 338).**

| Knowledge items | Correct response | Number | % |
|---|---|---|---|
| **1.** Healthcare workers are recommended for influenza vaccination in Thailand by the MOPH | True | 302 | 89.3 |
| **2.** Thailand has a national policy to provide free vaccine to high-risk groups | True | 273 | 80.8 |
| **3.** Influenza can lead to serious complications, including pneumonia | True | 240 | 71.0 |
| **4.** HCWs are less susceptible to influenza infections than other people | False | 219 | 64.8 |
| **5.** Influenza vaccination may not work if the vaccine contains the wrong mix of viruses | True | 176 | 52.1 |
| **6.** Influenza vaccination does not work in some persons, even if the vaccine has the right mix of viruses | True | 140 | 41.4 |
| **7.** Influenza vaccine may cause some people to get influenza | False | 102 | 30.2 |
| **Overall knowledge categories** | | | |
| Need improvement (Overall score <60%) | 0–3 | 104 | 30.8 |
| Fair (Overall score 60–79%) | 4–5 | 149 | 44.1 |
| Good (Overall score ≥80%) | 6–7 | 85 | 25.1 |

influenza vaccine does not itself cause some people to get influenza (Table 2). Overall, only a quarter of the participants had a good knowledge regarding influenza and vaccination (85; 25.1%) (Table 2).

## Influenza vaccine hesitancy among the participants

A majority of the participants reported some degree of hesitancy towards influenza vaccines (197; 58.3%), and 7 HCWs (2.1%) were outright refusers of influenza vaccines who were not hesitant in their refusal (Fig 1). On closer inspection, most of the hesitant HCWs were "hesitant compliers" who had doubts about the influenza vaccines but still accepted them (172; 87.3%). Only 3 (1.5%) refused the vaccine while being considerate of its benefits, and 22 (11.2%) delayed receiving the vaccine for themselves (Fig 2). More than 60% of the health management staff, other medical professionals (paramedics), and nurses reported vaccine hesitancy, while other HCW types had less hesitancy than the sample prevalence of 58.3%; the least hesitancy was seen among pharmacists (31.6%) (Fig 3).

## Vaccine hesitancy determinants

Removing the seven HCWs who refused influenza vaccines left with an analytical sample of 331 HCWs. The aggregated mean scores and standard deviations of the vaccine hesitancy determinants are presented in Table 3. The mean scores are on a scale of 1–5, where higher scores signify higher hesitancy or more negative sentiment towards the construct. Overall, the participants reported the highest negative perceptions towards pharmaceutical industry influences (mean score 2.8, SD 1.0) and influence of media/ social media (mean score 2.7, SD 0.8) (Table 3). When the mean scores were compared, HCW with influenza vaccine hesitancy were more likely to have a higher lack of trust in the health system (2.7 vs 2.3; p-value 0.008), negative past experience of vaccination (2.4 vs 2.1; p-value 0.017), negative perception of vaccination as a social norm (2.3 vs 2.0; p-value 0.011), negative influence of other HCW (2.2 vs 1.9; p-value 0.006), higher perceived risk and lack of benefit of influenza vaccine (2.4 vs 2.1; p-value 0.001), risk of adverse events due to vaccination (2.7 vs 2.3; p-value 0.001), fear of painful injections (2.5 vs 2.2; p-value 0.007), and lack of access to vaccines (2.5 vs 2.2; p-value <0.001) (Table 3).

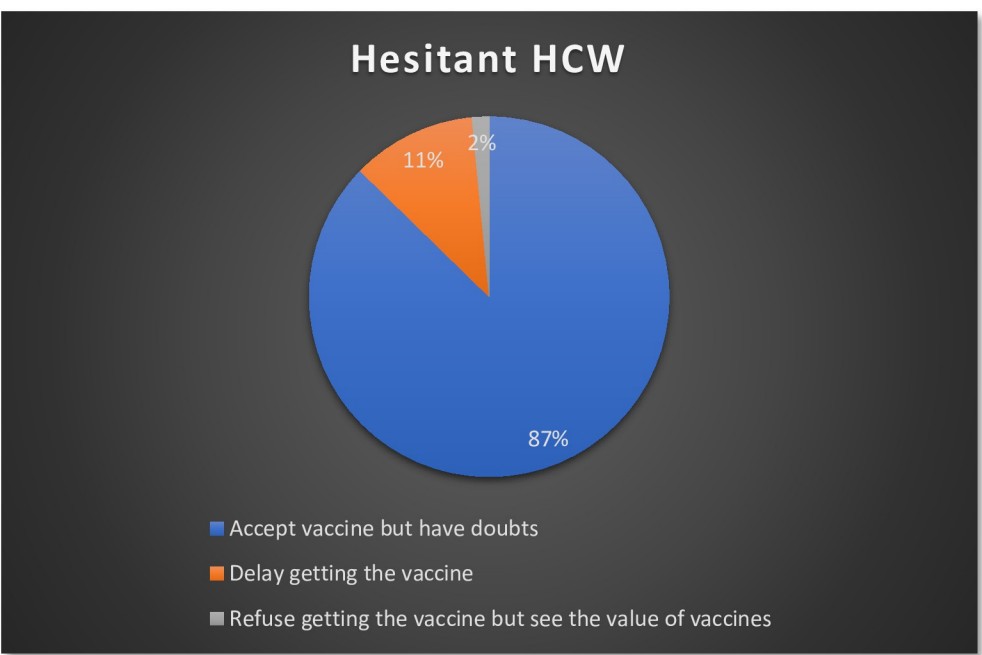

**Fig 2. Types of hesitant HCW among the participants.**

## Factors associated with influenza vaccine hesitancy

In a series of univariable logistic regression analyses, different variables were found to be significantly associated with influenza vaccine hesitancy (Table 4). Compared to young HCWs (aged 30 years and below), HCWs aged above 50 years were nearly three times more likely to be vaccine-hesitant (OR 2.8; 95% CI 1.2–6.3). Similarly, HCWs with 20 years or more work experience were more than two times more likely to be vaccine-hesitant than HCWs with

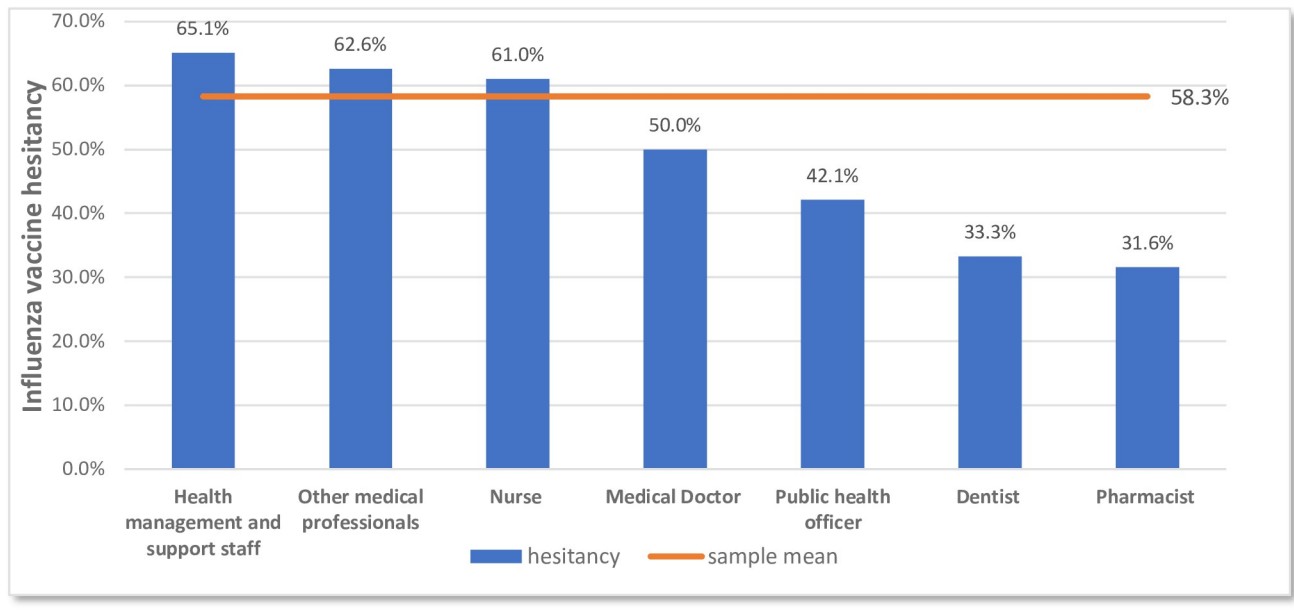

**Fig 3. Influenza vaccine hesitancy levels among different HCW types.**

Table 3. Influenza vaccine hesitancy determinants aggregated mean scores (n = 331).

| Determinant | Total Mean (SD) | Influenza vaccine hesitancy | | | | p-value* |
|---|---|---|---|---|---|---|
| | | Yes (n = 197) | | No (n = 134) | | |
| | | Mean | SD | Mean | SD | |
| **Contextual factors** | | | | | | |
| Negative influence of media/ social media | 2.7 (0.8) | 2.6 | 0.8 | 2.8 | 0.9 | 0.154 |
| Negative perception of politics/policies | 1.9 (0.9) | 1.9 | 0.8 | 1.8 | 0.9 | 0.057 |
| Negative perception of pharmaceutical industry | 2.8 (1.0) | 2.8 | 0.9 | 2.7 | 1.1 | 0.435 |
| Lack of trust in health system | 2.5 (1.0) | 2.7 | 0.7 | 2.3 | 1.1 | 0.008 |
| **Individual/Group influences** | | | | | | |
| Negative past experience of vaccination | 2.3 (1.1) | 2.4 | 1.1 | 2.1 | 1.0 | 0.017 |
| Negative beliefs, attitudes about influenza and vaccination | 2.2 (0.9) | 2.2 | 0.8 | 2.1 | 1.0 | 0.614 |
| Negative perception of vaccination as a social norm | 2.2 (0.9) | 2.3 | 0.8 | 2.0 | 0.9 | 0.011 |
| Negative influence of other HCW | 2.0 (1.0) | 2.2 | 0.9 | 1.9 | 1.0 | 0.006 |
| **Influenza vaccine specific factors** | | | | | | |
| Perceived risk of influenza infection and lack of benefit of vaccination | 2.2 (0.7) | 2.4 | 0.6 | 2.1 | 0.7 | 0.001 |
| Risk of adverse events due to vaccination | 2.5 (0.9) | 2.7 | 0.9 | 2.3 | 1.0 | 0.001 |
| Fear of painful injections | 2.4 (0.7) | 2.5 | 0.8 | 2.2 | 0.8 | 0.007 |
| Access to vaccines in the hospital | 2.4 (0.7) | 2.5 | 0.7 | 2.2 | 0.7 | <0.001 |

Abbreviations: SD, Standard deviation; Min, Minimum; Max, Maximum.

*p-value obtained from independent t-test.

experience of 5 years or less (OR 2.2; 95% CI 1.2–4.2). In contrast, HCWs having fair knowledge of influenza and vaccines were less likely to be hesitant compared to HCWs with less knowledge (OR 0.5; 95% CI 0.3–0.8).

Among the contextual factors, HCWs with negative perceptions of politics/policies and lack of trust in the health system had significant odds of being vaccine-hesitant compared to HCWs with positive perceptions (Table 4). Similarly, in the individual/group influences, significant determinants of hesitancy were negative past experience of vaccination (High—OR 2.1, 95% CI 1.2–3.7; Moderate–OR 1.7, 95% CI 1.0–2.9), moderate level of negative beliefs (OR 2.0, 95% CI 1.2–3.5), high negative perception of vaccination as a social norm (OR 2.2, 95% CI 1.3–3.9), and negative influence of other HCW (High–OR 3.3, 95% CI 1.9–5.7; Moderate—OR 2.7, 95% CI 1.5–4.9). Among the factors specific to influenza vaccines, low perceived risk (Low—OR 2.5, 95% CI 1.4–4.45; Moderate–OR 1.9, 95% CI 1.1–3.2), high risk of adverse events after vaccination (OR 2.8, 95% CI 1.5–5.3), fear of painful injections (High–OR 1.7, 95% CI 1.1–2.7; Moderate—OR 2.3, 95% CI 1.0–4.8), and reduced access to vaccines in the hospital (Low–OR 2.0, 95% CI 1.2–3.5; Moderate–OR 2.1, 95% CI 1.2–3.5) were significantly associated with the hesitancy.

However, in the multivariable model, only age (above 50 years–aOR 3.2, 95% CI 1.3–8.5), fair knowledge (aOR 0.4, 95% CI 0.2–0.8), and negative influence of other HCW (High–aOR 2.3, 95% CI 1.1–4.8; Moderate—aOR 2.1, 95% CI 1.1–4.4) remained significant factors of influenza vaccine hesitancy (Table 4). Work experience was not entered in the final model as it showed a high correlation with age.

## Discussion

Influenza vaccine hesitancy was highly prevalent among the Thai HCWs in this study, as nearly 60% of the survey participants reported having some degree of hesitancy. Older age and

**Table 4. Factors associated with vaccine hesitancy among HCW (n = 331).**

| Factor | Influenza vaccine hesitancy | | Crude OR | 95% CI | Adjusted OR[a] | 95% CI |
|---|---|---|---|---|---|---|
| | Yes (n = 197) | No (n = 134) | | | | |
| | n (%) | n (%) | | | | |
| Age-groups (in years) | | | | | | |
| ≤30 | 55 (53.9) | 47 (46.1) | Ref | | Ref | |
| 31–40 | 65 (55.1) | 53 (44.9) | 1.0 | 0.6–1.8 | 1.1 | 0.6–2.1 |
| 41–50 | 44 (64.7) | 24 (35.3) | 1.6 | 0.8–2.9 | 1.9 | 0.9–4.1 |
| >50 | 33 (76.7) | 10 (23.3) | 2.8 | 1.2–6.3* | 3.2 | 1.3–8.5* |
| Sex | | | | | | |
| Male | 34 (59.6) | 23 (40.4) | Ref | | | |
| Female | 163 (59.5) | 111 (40.5) | 0.9 | 0.6–1.8 | | |
| Level of hospital | | | | | | |
| Provincial | 54 (55.7) | 43 (44.3) | Ref | | | |
| District | 143 (61.1) | 91 (38.9) | 1.2 | 0.8–2.0 | | |
| HCW type | | | | | | |
| Health service providers | 169 (58.3) | 121 (41.7) | 0.6 | 0.3–1.3 | | |
| Health management and support staff | 28 (68.3) | 13 (31.7) | Ref | | | |
| Average daily patient exposure | | | | | | |
| High | 96 (56.8) | 73 (43.2) | 0.8 | 0.4–1.5 | | |
| Moderate | 65 (63.1) | 38 (36.9) | 1.1 | 0.6–2.1 | | |
| Low | 36 (61.0) | 23 (39.0) | Ref | | | |
| Work experience | | | | | | |
| ≤5 years | 37 (50.0) | 37 (50.0) | Ref | | | |
| >5 years to <20 years | 95 (58.3) | 68 (41.7) | 1.4 | 0.8–2.4 | | |
| ≥20 years | 65 (69.1) | 29 (30.9) | 2.2 | 1.2–4.2* | | |
| Highest educational attainment | | | | | | |
| High school or below | 25 (69.4) | 11 (30.6) | Ref | | | |
| Diploma/ vocational college | 31 (72.1) | 12 (27.9) | 1.1 | 0.4–3.0 | | |
| Bachelor's degree | 122 (56.2) | 95 (43.8) | 0.6 | 0.3–1.2 | | |
| Master's degree or higher | 19 (54.3) | 16 (45.7) | 0.5 | 0.2–1.4 | | |
| Pre-existing chronic condition | | | | | | |
| Yes | 44 (60.3) | 29 (29.7) | 1.0 | 0.6–1.8 | | |
| No | 153 (59.3) | 105 (40.7) | Ref | | | |
| Knowledge of influenza and vaccination | | | | | | |
| Need improvement | 71 (69.6) | 31 (30.4) | Ref | | Ref | |
| Fair | 74 (51.4) | 70 (48.6) | 0.5 | 0.3–0.8* | 0.4 | 0.2–0.8* |
| Good | 52 (61.2) | 33 (38.8) | 0.7 | 0.4–1.3 | 0.9 | 0.5–2.0 |
| **Contextual factors** | | | | | | |
| Negative influence of media/ social media | | | | | | |
| High | 41 (57.7) | 30 (42.3) | 0.7 | 0.4–1.4 | | |
| Moderate | 93 (57.1) | 70 (42.9) | 0.7 | 0.4–1.2 | | |
| Low | 63 (64.9) | 34 (35.1) | Ref | | | |
| Negative perception of politics/policies | | | | | | |
| High | 76 (69.1) | 34 (30.9) | 2.6 | 1.5–4.7* | 1.8 | 0.8–4.0 |
| Moderate | 78 (61.4) | 49 (38.6) | 1.9 | 1.1–3.2* | 1.2 | 0.6–2.3 |
| Low | 43 (45.7) | 51 (54.3) | Ref | | Ref | |
| Negative perception of pharmaceutical industry | | | | | | |
| High | 58 (65.9) | 30 (34.1) | 1.6 | 0.9–2.9 | | |

*(Continued)*

**Table 4.** (Continued)

| Factor | Influenza vaccine hesitancy | | Crude OR | 95% CI | Adjusted OR[a] | 95% CI |
|---|---|---|---|---|---|---|
| | Yes (n = 197) | No (n = 134) | | | | |
| | n (%) | n (%) | | | | |
| Moderate | 89 (58.6) | 63 (41.4) | 1.1 | 0.7–1.9 | | |
| Low | 50 (54.9) | 41 (45.1) | Ref | | | |
| Lack of trust in health system | | | | | | |
| High | 52 (69.3) | 23 (30.7) | 2.6 | 1.3–4.9* | 1.3 | 0.5–3.5 |
| Moderate | 106 (61.3) | 67 (38.7) | 1.8 | 1.1–3.0* | 1.2 | 0.6–2.4 |
| Low | 39 (47.0) | 44 (53.0) | Ref | | Ref | |
| **Individual/group factors** | | | | | | |
| Negative past experience of vaccination | | | | | | |
| High | 65 (67.0) | 32 (33.0) | 2.1 | 1.2–3.7* | 1.4 | 0.6–3.2 |
| Moderate | 80 (62.5) | 48 (37.5) | 1.7 | 1.0–2.9* | 1.2 | 0.6–2.5 |
| Low | 52 (49.1) | 54 (50.9) | Ref | | Ref | |
| Negative beliefs, attitudes about influenza and vaccination | | | | | | |
| High | 62 (59.0) | 43 (41.0) | 1.4 | 0.8–2.5 | 0.7 | 0.3–1.5 |
| Moderate | 86 (67.2) | 42 (32.8) | 2.0 | 1.2–3.5* | 1.4 | 0.7–2.7 |
| Low | 49 (50.0) | 49 (50.0) | Ref | | Ref | |
| Negative perception of vaccination as a social norm | | | | | | |
| High | 78 (68.4) | 36 (31.6) | 2.2 | 1.3–3.9* | 0.9 | 0.4–2.3 |
| Moderate | 70 (59.3) | 48 (40.7) | 1.5 | 0.9–2.5 | 1.1 | 0.5–2.2 |
| Low | 49 (49.5) | 50 (50.5) | Ref | | Ref | |
| Negative influence of other HCW | | | | | | |
| High | 92 (70.2) | 39 (29.8) | 3.3 | 1.9–5.7* | 2.3 | 1.1–4.8* |
| Moderate | 60 (65.9) | 31 (34.1) | 2.7 | 1.5–4.9* | 2.1 | 1.1–4.4* |
| Low | 45 (41.3) | 64 (58.7) | Ref | | Ref | |
| **Influenza vaccination specific factors** | | | | | | |
| Perceived risk of influenza infection and lack of benefit of vaccination | | | | | | |
| Low | 73 (68.2) | 34 (31.8) | 2.5 | 1.4–4.5* | 1.0 | 0.4–2.5 |
| Moderate | 82 (61.7) | 51 (38.3) | 1.9 | 1.1–3.2* | 0.9 | 0.4–1.8 |
| High | 42 (46.2) | 49 (53.8) | Ref | | Ref | |
| Risk of adverse events due to vaccination | | | | | | |
| High | 55 (73.3) | 20 (26.7) | 2.8 | 1.5–5.3* | 1.6 | 0.6–4.1 |
| Moderate | 84 (60.9) | 54 (39.1) | 1.6 | 0.9–2.6 | 0.9 | 0.5–1.9 |
| Low | 58 (49.2) | 60 (50.8) | Ref | | Ref | |
| Fear of painful injections | | | | | | |
| High | 104 (63.8) | 59 (36.2) | 1.7 | 1.1–2.7* | 0.9 | 0.5–1.6 |
| Moderate | 28 (70.0) | 12 (30.0) | 2.3 | 1.0–4.8* | 1.8 | 0.7–4.5 |
| Low | 65 (50.8) | 63 (49.2) | Ref | | Ref | |
| Access to vaccines in the hospital | | | | | | |
| Low | 64 (66.0) | 33 (34.0) | 2.0 | 1.2–3.5* | 1.3 | 0.6–2.8 |
| Moderate | 71 (66.4) | 36 (33.6) | 2.1 | 1.2–3.5* | 1.4 | 0.7–2.8 |
| High | 62 (48.8) | 65 (51.2) | Ref | | Ref | |

Abbreviations: OR, Odds Ratio; CI, Confidence Interval

[a]Adjusted OR and 95% CI, and p-values calculated using multivariable logistic regression, controlling for hospital locations to account for clustering

the role of other HCWs were associated with increased hesitancy, while fair knowledge of influenza and vaccination was indicated to reduce hesitancy. The findings of this study can be used to improve local influenza vaccination policy in Thailand.

Since vaccine hesitancy studies are in their nascent stage in Thailand, there are no prior estimates of the prevalence of influenza vaccine hesitancy among the HCWs. Therefore, the proportion of 58.3% found in this study can function as a baseline figure of influenza vaccine hesitancy among Thai HCWs for future reference. Nevertheless, comparisons can be made with past Thai studies regarding the HCW doubts of the influenza vaccines. Nearly 50% of HCWs had some doubts about the influenza vaccine in this current study, which is higher than the 30% reported among Thai physicians in 2017 [29]. In a similar vein, the vaccine acceptance rate in this study was lower than the 65.4% reported among HCWs in a Bangkok hospital [30]. Although these comparisons are limited due to differences in vaccine hesitancy conception, study sites, and the HCW definition, which was expanded to include all types of HCW working in the hospital in this study as opposed to only doctors or medical service providers in the past studies [29, 30], they indicate that HCW in Nakhon Phanom may be more hesitant towards influenza vaccines than normally expected. Our result can be comparable to the 65% influenza vaccine hesitancy reported among community health workers in Southwest China, where, similar to our study, older and complacent HCWs were more likely to be vaccine-hesitant [41]. Nonetheless, similar studies from other countries, such as Egypt, Hong Kong, and South Africa, report influenza vaccine hesitancy among HCWs to be below 50% [42–45].

While the proportion of hesitant HCWs in this study is alarming, the proportion of HCWs who actively delayed and refused vaccination was only around 11%. Most of the hesitant HCWs were those who were skeptical of the influenza vaccine but still received the vaccination. A possible explanation for this phenomenon may be that although there isn't a mandatory vaccination for HCWs and there is a notion of HCW's choice, in reality, the HCWs often end up taking the influenza vaccines as an obligation, particularly as Nakhon Phanom is a "model" province for influenza vaccine research in Thailand where vaccination rates are supposed to be higher.

In this study, the role of other HCWs, such as the medical doctor, vaccinator nurses, or senior staff, was a significant predictor of influenza vaccine hesitancy. This is an important finding as it suggests that HCW vaccine hesitancy can be potentially reduced when HCWs, such as medical doctors and supervisors, provide positive feedback and peer support to their fellow HCWs. Medical doctors have a higher social class [46], and along with vaccinator nurses, senior HCWs have more social capital to be influential in social interactions with other HCWs in hospital settings [47]. As many studies indicate, the acceptance of influenza vaccination among this group is likely to be transferred to their HCW colleagues. For example, supervisor and physician encouragement has been found to be a key predictor of HCW vaccination [48]. The strategy of using influential HCWs and senior staff was found to improve HCW vaccination in research conducted in countries such as Spain and Canada [49, 50]. Similarly, in Israel, a randomized controlled trial showed that an intervention comprising of a lecture from a family physician, in addition to e-mail reminders, and a personal approach from a key local staff (doctor or nurse), was able to increase the odds of influenza vaccination among HCW by 3.51 (95% CI 2.03–6.09) compared to the control group [51]. Our study also provides supportive evidence that increasing knowledge of influenza disease and vaccine may help to reduce hesitancy to a certain extent, particularly when combined with support from an influential senior HCW.

The social capital of senior HCWs (i.e., the capital derived from one's social position and status) may explain why the vaccine-hesitant HCWs were more likely to be older than 50 years

in our study. Using Bourdieu's concept of social capital [52], it can be hypothesized that when this group of experienced and senior HCWs becomes influenza vaccine-hesitant, they can exert their power to either resist (i.e., refuse) or negotiate (i.e., delay) taking vaccines. Hesitant HCWs of this group may have more options instead of being confined to the government vaccines and thus have more power to choose. In previous research, vaccine hesitancy was viewed as a form of symbolic capital that helped to understand and elaborate on the social processes, uncertainties, and difficulties in various vaccine-related journeys [53]. While in this study, the older HCW's accrued social capital with the virtue of their seniority and social networks built in the hospital for many years may have provided them with more power to resist the vaccination policy and the approaches of junior vaccinators. The higher social standing of the older HCWs may be the reason why they could overtly display their hesitancy, while the younger HCWs may have had to keep it hidden. This hypothesis, however, needs to be further explored and corroborated by other studies.

Overall, this study contributes to the growing literature on vaccine hesitancy and presents the situation of HCWs in a rural northeastern province in Thailand. This study has been reported here in accordance with the STROBE statement on cross-sectional studies (S3 Table). Nevertheless, there are some limitations in the study. First, the research was conducted after the first wave of coronavirus disease 2019 (COVID-19) in Thailand. This presented different logistic hurdles and time delays in conducting the survey. More importantly, the emergence of COVID-19 might have affected some HCWs' influenza vaccine decision-making, which was not captured in the survey data as the questionnaire was designed in the pre-COVID period. Second, there may have been a potential selection bias as the data was collected exclusively in the daytime, and the HCWs working night shifts were unable to be included. Although stratified random sampling was used in the survey, which accounted for all staff working in the hospital and gave the HCW working at night time an equal probability to get enrolled, cross-sectional data collection in the daytime may have limited their participation. Finally, a small number of samples in some HCW categories reduced the statistical power of the survey to draw valid comparisons between the HCW types. Nonetheless, the survey provided a representative sample of HCWs of the province selected by a multistage cluster sampling using probability proportional to HCW size in the hospitals.

In conclusion, a high proportion of HCWs had some form of hesitancy towards influenza vaccines. The findings of this study can be used as a baseline to monitor the changes in the level of vaccine hesitancy of Thai HCWs in future studies. Public health strategies and interventions informed by the study findings may be more effective in reducing influenza vaccine hesitancy among Thai HCW.

## Supporting information

**S1 Table. HCW influenza vaccine hesitancy determinant items (5 point Likert scale).**
(PDF)

**S2 Table. Response rates from each of the study hospitals.**
(PDF)

**S3 Table. STROBE statement—Checklist of items that should be included in reports of cross-sectional studies.**
(PDF)

**S1 File. Influenza vaccine coverage by province.**
(XLSX)

## Acknowledgments

We are grateful to the HCWs who participated in the study and the focal points at the six hospitals who helped in conducting the survey.

## Author Contributions

**Conceptualization:** Manash Shrestha, Penchan Pradubmook Sherer, Seung Chun Paek, Prabda Praphasiri.

**Data curation:** Manash Shrestha, Kriengkrai Prasert, Sutthinan Chawalchitiporn.

**Formal analysis:** Manash Shrestha, Seung Chun Paek, Kriengkrai Prasert.

**Investigation:** Manash Shrestha, Penchan Pradubmook Sherer, Seung Chun Paek, Prabda Praphasiri.

**Methodology:** Manash Shrestha, Penchan Pradubmook Sherer, Seung Chun Paek, Prabda Praphasiri.

**Project administration:** Manash Shrestha, Kriengkrai Prasert, Sutthinan Chawalchitiporn.

**Resources:** Kriengkrai Prasert, Prabda Praphasiri.

**Supervision:** Penchan Pradubmook Sherer, Seung Chun Paek, Prabda Praphasiri.

**Validation:** Penchan Pradubmook Sherer, Seung Chun Paek.

**Visualization:** Manash Shrestha, Seung Chun Paek, Prabda Praphasiri.

**Writing – original draft:** Manash Shrestha.

**Writing – review & editing:** Manash Shrestha, Penchan Pradubmook Sherer, Seung Chun Paek, Kriengkrai Prasert, Sutthinan Chawalchitiporn, Prabda Praphasiri.

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
