## [Decision Letter · Decision Letter 0]

1 May 2024

PONE-D-24-10541Influenza vaccine hesitancy among healthcare workers in a Northeastern province in Thailand: Findings of a cross-sectional surveyPLOS ONE

Dear Dr. Praphasiri,

Thank you for submitting your manuscript to PLOS ONE. After careful consideration, we feel that it has merit but does not fully meet PLOS ONE’s publication criteria as it currently stands. Therefore, we invite you to submit a revised version of the manuscript that addresses the points raised during the review process.

We look forward to receiving your revised manuscript.

Kind regards,

Fadi Aljamaan

Academic Editor

PLOS ONE

Journal Requirements:

Additional Editor Comments:

Dear authors

Please respond to the comments of reviewer 2 who raised many major points in methodology and results interpretation before we progress to second cycle of revision, regarding reviewer 1 I am contacting him whether he meant rejection or no as he did not mention clear reasons of rejection.

Reviewers' comments:

Reviewer's Responses to Questions

**Comments to the Author**

1. Is the manuscript technically sound, and do the data support the conclusions?

Reviewer #1: Yes

Reviewer #2: Partly

2. Has the statistical analysis been performed appropriately and rigorously? 

Reviewer #1: Yes

Reviewer #2: No

3. Have the authors made all data underlying the findings in their manuscript fully available?

Reviewer #1: Yes

Reviewer #2: No

4. Is the manuscript presented in an intelligible fashion and written in standard English?

Reviewer #1: Yes

Reviewer #2: Yes

5. Review Comments to the Author

Reviewer #1: thank you for submitting this work titled Influenza vaccine hesitancy among healthcare workers in a Northeastern province in Thailand: Findings of a cross-sectional survey which Background: Healthcare workers (HCWs) are an important target group for influenza

36 vaccination due to their increased risk of infection. However, their uptake remains a challenge.

37 This study aimed to identify and measure influenza vaccine hesitancy among HCWs in Nakhon

38 Phanom province, Thailand.

39

40 Methods: A representative cross-sectional survey was conducted during August – September

41 2020, among 350 HCWs at six hospitals in the province selected by a two-stage cluster

42 sampling using a self-administered questionnaire. HCWs who either delayed getting influenza

43 vaccines, accepted the vaccines but were unsure, or refused the vaccine with doubts were

44 categorized as hesitant. HCWs who accepted the influenza vaccine without any doubts were

45 classified as non-hesitant. Determinants of vaccine hesitancy were identified by a multivariable

46 logistic regression analysis.

47

48 Results: A total of 338 participants (97%) filled the questionnaires. The mean age of the

49 participants was 37.2 years. Most participants were female (280; 83%), nurses (136; 40%),

50 working at district hospitals (238; 71%), with bachelor’s degree (223; 66%), and without any

51 pre-existing chronic medical conditions (264; 78%). Influenza vaccine hesitancy was evident

52 among nearly 60% of the participants (197/338), who had varying patterns of hesitancy.

53 Significant factors of influenza vaccine hesitancy were found to be age above 50 years

54 (adjusted odds ratio [aOR] 3.2, 95% CI 1.3-8.5), fair knowledge of influenza and vaccination

55 (aOR 0.4, 95% CI 0.2-0.8), and negative influence of other HCW (High level– aOR 2.3, 95%

56 CI 1.1-4.8; Moderate level- aOR 2.1, 95% CI 1.1-4.4).

57

58 Conclusion: Influenza vaccine hesitancy was highly prevalent among the Thai HCWs in

59 Nakhon Phanom province. Imparting updated information to the HCW, in combination with

60 positive guidance from influential HCWs in the hospital, may help reduce hesitancy. These

61 data may be useful to the National Immunization Program to design appropriate approaches to

Reviewer #2: Thank you for this important work. Please, find my comments below

1. On what basis did the authors include those who “accepted the vaccines but were unsure” as HESITANTS. Can you cite any reference/guideline/consensus statement for this? And what is the actual number of the study participants who fall into this category? Lumping those who indicated acceptance but were unsure cannot be accurate since the concept of vaccine acceptance/hesitancy itself is dynamic being determined by many other external influences. And I think this may be the reason why only 3 factors were significantly associated with hesitancy in the final adjusted model, with the age factor being an odd finding with respect to the overwhelming findings of previous studies.

2. Can the authors specifically define what they mean by “acceptance”, whether, for example, the term includes those already vaccinated plus unvaccinated who are willing to be vaccinated, or whether the term refers only to unvaccinated who are willing to be vaccinated?

3. Can the authors provide a reference for their affirmative statement “…acceptance of vaccination by HCWs themselves is, therefore, also an important determinant of influenza vaccine uptake in the populations that they serve”

4. Please cite the webpage/URL for your statement in line 101-103, “However, in 2018, regional reports showed that HCW influenza vaccine coverage had plummeted to below 30% in some regions (Department of Medical Services [DMS], MOPH)”.

5. If the “mixed-methods study”, which the present study is said to be part of by the authors has been published, please cite it were you said this.

6. Why would the authors say “As there was no prior estimate of influenza vaccine hesitancy among Thai HCWs” in line 160 under Methods section, while in line of the introduction, they wrote “A study 105 revealed that only 60% of Thai HCWs were willing to receive the pandemic influenza (H1N1) vaccine [21]”? Isn’t this a self-contradiction?

7. In your methods, please describe the tool used to assessed knowledge, specify whether it is adapted or adopted and whether it has been validated.

8. In the analysis, have the authors evaluated model fitness? If so, please state which were used, and if not, please explain why.

9. Given that this study was conducted during the early phase of the COVID-19 pandemic, a period that has saw an unprecedented levels of rise in vaccine hesitancy and antagonism, have the authors considered a spillover of this COVID-19 vaccine-related hesitancy into influenza vaccine and all other vaccines?

10. In your Discussion section, please clarify why you would say that there are “no prior estimates of the prevalence of influenza vaccine hesitancy among the HCWs” despite citing acceptance rates of the same vaccine among the same study population in the same country. Note subtracting these reported acceptance rates gives you the hesitancy rates, right?

11. The probable reason(s) for higher odds of hesitancy in HCWs age >50years (who at higher risk of the infection than those below) need to be better hypothesized in your Discussion. I think this finding may be related to the inaccurate analysis that lumps “those who accept but were unsure” as hesitants.

I look forward to reading the revised version of this manuscript.

Best

6. PLOS authors have the option to publish the peer review history of their article (what does this mean?). If published, this will include your full peer review and any attached files.

Reviewer #1: No

Reviewer #2: **Yes: **Dr. Sahabi Kabir Sulaiman

---

## [Author Response · Author response to Decision Letter 0]

14 Jun 2024

A rebuttal letter in the form of a response to reviewers has been submitted as a Word file, and appropriate changes have been made to the manuscript.

---

## [Decision Letter · Decision Letter 1]

27 Aug 2024

PONE-D-24-10541R1Influenza vaccine hesitancy among healthcare workers in a Northeastern province in Thailand: Findings of a cross-sectional surveyPLOS ONE

Dear Dr. Praphasiri,

Thank you for submitting your manuscript to PLOS ONE. After careful consideration, we feel that it has merit but does not fully meet PLOS ONE’s publication criteria as it currently stands. Therefore, we invite you to submit a revised version of the manuscript that addresses the points raised during the review process.

We look forward to receiving your revised manuscript.

Kind regards,

Sirwan Khalid Ahmed

Academic Editor

PLOS ONE

Journal Requirements:

Reviewers' comments:

Reviewer's Responses to Questions

**Comments to the Author**

1. If the authors have adequately addressed your comments raised in a previous round of review and you feel that this manuscript is now acceptable for publication, you may indicate that here to bypass the “Comments to the Author” section, enter your conflict of interest statement in the “Confidential to Editor” section, and submit your "Accept" recommendation.

Reviewer #2: All comments have been addressed

Reviewer #3: (No Response)

2. Is the manuscript technically sound, and do the data support the conclusions?

Reviewer #2: Yes

Reviewer #3: Yes

3. Has the statistical analysis been performed appropriately and rigorously? 

Reviewer #2: Yes

Reviewer #3: Yes

4. Have the authors made all data underlying the findings in their manuscript fully available?

Reviewer #2: (No Response)

Reviewer #3: Yes

5. Is the manuscript presented in an intelligible fashion and written in standard English?

Reviewer #2: Yes

Reviewer #3: Yes

6. Review Comments to the Author

Reviewer #2: Thank you for revising this work. I have gone through the revision and its supporting responses and I agree with them. I have no new comments. Looking forward to seeing the published work online.

Best

Reviewer #3: 2. How the author makes sure that the participants were not biased towards specific answer

3. How the author makes sure that the participants were not disengaged while answering.

4. How the author makes sure about reliability of result

5. Author need to mention reliability and validity of tools used in the study

6. Author need to mention response rate

7. PLOS authors have the option to publish the peer review history of their article (what does this mean?). If published, this will include your full peer review and any attached files.

Reviewer #2: **Yes: **Sahabi Kabir Sulaiman

Reviewer #3: No

---

## [Author Response · Author response to Decision Letter 1]

4 Sep 2024

The rebuttal letter has been added in the attached files as "Response to Reviewers".

---

## [Editor Report · Decision Letter 2]

6 Sep 2024

Influenza vaccine hesitancy among healthcare workers in a Northeastern province in Thailand: Findings of a cross-sectional survey

PONE-D-24-10541R2

Dear Dr. Praphasiri,

We’re pleased to inform you that your manuscript has been judged scientifically suitable for publication and will be formally accepted for publication once it meets all outstanding technical requirements.

Kind regards,

Sirwan Khalid Ahmed

Academic Editor

PLOS ONE
---

## [Editor Report · Acceptance letter]

11 Sep 2024

PONE-D-24-10541R2 

PLOS ONE

Dear Dr. Praphasiri, 

I'm pleased to inform you that your manuscript has been deemed suitable for publication in PLOS ONE. Congratulations! Your manuscript is now being handed over to our production team.

Kind regards, 

on behalf of

Dr. Sirwan Khalid Ahmed 

Academic Editor

PLOS ONE